# Anishinabek Women's Nibi Giikendaaswin (Water Knowledge)

**Susan Chiblow (Ogamauh annag qwe)**

Faculty of Environmental Studies (FES), York University, Toronto, ON M3J 1P3, Canada; suechiblow@mississaugi.com; Tel.: +1-(705)-975-1604

**Abstract:** This paper springs from conversations and my life experiences with Anishinabek Elders and practitioners, which includes my understanding of my life journey in re-searching for Anishinabe qwe (woman) giikendaaswin (knowledge, information, and the synthesis of our personal teachings). Anishinabek women have giikendaaswin about nibi (water) that can transform nibi (water) governance. Re-searching for giikendaaswin is directly linked to inclusive decision-making. This paper describes how Anishinabek understand and construct giikendaaswin based on Anishinabek ontology and epistemology, which includes nibi (water) giikendaaswin. This supports what Anishinabek know, how we come to know, how we know it to be true, and ultimately why we seek giikendaaswin.

**Keywords:** Anishinabek; nibi (water); women; governance; giikendaaswin

---

Nibi is one of the Anishinabek words used for the waters. It is a general word referring to all waters, including lakes, rivers, streams, rain, etc., but each individual body of water including rain has another word in Anishinabemowin (Anishinabek language). Nibi is, therefore, considered to encompass all waters and many scholars now use the term "waters" to stress the importance of the different types of water. I use the term "nibi" and "nibi governance" when specifically referring to Anishinabek giikendaaswin and use the word "water" and "water governance" when used for referencing; however, I also use "waters" to stress the understanding of the different types of water.

## 1. Introduction

Some scholars question Anishinabek giikendaaswin (knowledge) as a stand-alone science [1] and its validity. Anishinabek peoples have lived on the lands for thousands of years, acquiring giikendaaswin and governing themselves based on their own giikendaaswin. "Indigenous peoples are the original researchers of these territories" [2]. Smith (1999) explains that the "development of western, rational, scientific knowledge was connected with colonialism and the 'discovery' of the Indigenous Others by Europeans" [3]. Similarly, Battiste (2002) indicated that knowledge production from the west constitutes a form of imperialism that disregards and erases other types of knowledge [4]. Attempting to erase the knowledge of other peoples has implications of not knowing the best sustainable practices for nibi. Kimmerer (2013) refers to western science being asleep at the wheel as rivers are contaminated [5] and, if we honestly do a brief inventory, the world is on the verge of collapse due to several factors including decision-making in nibi governance. Many activists, scientists, and Anishinabek peoples observe that water governance in Canada is in a state of crisis [6].

Anishinabek giikendaaswin can also be referred to as traditional ecological knowledge (TEK). Extensive literature was written on Indigenous knowledge since the 1980s [1]. This type of knowledge is cumulative, holistic, experiential, and dynamic. McGregor (2013) stated that it is fundamentally about relationships which includes all our ancestors, all living things, the spirit world, and those yet to come [7]. Relationships with and responsibilities to nibi are important in nibi governance.

Here, I argue for responsibility-based governance based on Anishinabek giikendaaswin which is about relationships and responsibilities. Kermoal and Altamirano-Jimenez (2016) explained that knowledge can be interrelated with territory, kinship, identity, governance, economy, and education [1]. They further elaborated that this knowledge can be crucial in managing natural resources, but it is imperative to avoid and essentialize stereotypical pan-Indigenous ways of knowing, as Anishinabek have a unique set of giikendaaswin situated to their sense of place [1]. Anishinabek Elders pursue giikendaaswin as lifelong learning willing to share for the protection of nibi. Absolon (2011) explained that "Indigenous knowledge occupies itself with the past, the present, and the future" [8], which covers the lifelong journey of learning. Indigenous knowledge comes from ancestral teachings and is as old as life; if one can listen to all life, it can transport us across planes of existence [8]. Understanding the past conditions of nibi will assist with the present to bring us into a healthy nibi future. Bell (2013) stated that Indigenous knowledge embodies the principles of sustainability [9]. Truly being sustainable is imperative for those yet unborn.

Current colonial laws for nibi are failing all peoples and nibi itself by ignoring the giikendaaswin of Anishinabe peoples. In a conversation with Elder Willie Pine, he shared giikendaaswin about nibi, about a time when he remembered being able to dip a cup into the Mississaugi River and the in-land lakes to get a drink of nibi, but today that is not so as nibi is sick. The Water Declaration of the Anishinaabek, Mushkegowuk, and Onkwehonwe in Ontario has a section on the conditions of the waters, stating how the different lakes, rivers, oceans, and streams are poisoned by "foreign economic values" violating "our sacred laws given by the Creator" [10]. The interests of capital by corporations weaponized water causing contamination [11]. Bakker et al. (2018) discussed the historical inequalities related to accessing water and how Canada's fragmented systems of water governance contributed to increased risks in water supply systems [12]. Indigenous Elders across Ontario stated that there are several reasons why Indigenous knowledge is not included or implemented in environmental decision-making. These reasons range from the lack of meaningful consultation, the fact that different knowledge systems do not necessarily align, and from the lack of trust in providing Indigenous knowledge as it was misused in the past. This lack of implementing Indigenous knowledge into environmental decision-making caused challenges and gaps in water governance [13]. Simms et al. (2016) examined the imposition of colonial governance frameworks excluding Indigenous peoples and their knowledge in water governance [14]. They concluded that existing colonial governance is fragmented, excludes Indigenous laws and knowledge, is entrenched in colonial approaches, and lacks the capacity and funding for First Nation participation [14].

There is currently a lack of gender balance in water policies, strategies, and governance. Anderson et al. (2013) suggested the current approaches discount Indigenous women, ignoring a "valuable perspective on water that could help to identify new ways of managing water . . . " [15]. The continuous ignoring of Anishinabek women is embedded in colonialism, is historic, and persists today. Todd (2016) explained the barriers linked to colonialism such as the "Euro-American" institutions that bequeath the "Euro-Western" systems and people rather than Indigenous women [16]. The historic disregard for Anishinabek women's knowledge stems from the original settlers "exploring" the lands, which were observations and experiences of European men whose interactions were based on their cultural views of gender resonating the role of women in European societies [3]. These observations and experiences by white explorers displaced Anishinabek women by trying to erase their giikendaaswin systems and legal and political realities. Many Anishinabek women are re-establishing their relationships with and responsibilities to nibi through various means such as the Water Walks, Idle No More, and water ceremonies. An Elder once told me that it is the women who will make the necessary changes to stop the destruction to nibi and the lands, but will it be too late?

The intent of this paper is to provide my understanding of Anishinabek women's nibi governance through the lens of Anishinabek ontology and epistemology. In minobimadazwin (the good life), everything is connected. Elders in ceremony explained that giikendaaswin is all encompassing of knowledge of how to live based on beliefs, following instructions, and constantly participating in the

search for minobimadazwin. There is no divide between ontology and epistemology; in fact, trying to explain these terms to Elders or language speakers is the cause of much discussion and humor. Contrasting the Anishinabek understanding of ontology and epistemology, Yates et al. (2015) described ontology as the nature of being simply, meaning that what you believe and how you live those beliefs is the way of being [17]. Meyer (2004) stated that epistemology is studying knowledge and is the way of knowing [18]. My understanding from ceremony, Anishinabek Elders, and practitioners is that the way of knowing is the way of being. These definitions divide my understanding of what it means to be Anishinabe qwe in searching for minobimadazwin; thus, I hereafter hyphenate both words in an attempt to articulate what I am conveying. This understanding is my own pedagogy, which is based on sources of giikendaaswin from Anishinabek Elders, practitioners, ceremony, literature, my family, and my infancy understanding of Anishinabemowin (Ojibway language). The insights are my reflections based on my understandings.

In particular, I convey my understandings of Anishinabek ontology-epistemology utilizing Anishinabek Elders', ceremony, scholars', and practitioners' giikendaaswin to reinforce what I have come to understand from my own pedagogy. I engage and construct nibi giikendaaswin based on Anishinabek nibi ontology-epistemology by first providing a brief background and explaining what nibi is from an Anishinabek women's perspective, re-affirming the need for collaboration on nibi governance as, in many instances, Anishinabek women's giikendaaswin is not being utilized in current water decision-making regimes. Secondly, I explain what nibi governance means and the role of Anishinabek women in nibi governance, including the importance of understanding the responsibilities with and relationships to nibi that come with nibi giikendaaswin. I articulate how that responsibility to and relationship with nibi includes sharing what we understand in a respectful way, which can contribute to nibi management policies in nibi management strategies. I conclude with how space needs to be created for Anishinabek giikendaaswin focusing on responsibility-based governance.

This goal will provide insight into humanity's relationship with nibi from an Anishinabek perspective allowing Anishinbek ontology-epistemology concepts to re-emerge. The conveyance of Anishinabek perspectives on nibi will highlight Anishinabek nibi giikendaaswin, contrasting the politics of nibi and nibi governance of colonialism and the settler understanding of nibi.

## 2. Indigenous Research Methods and Positionality

My positionality drives my personal interest, and my being, as an Anishinabe qwe from the Great Lakes territory and, as such, I can only speak from my understandings as an Anishinabe qwe perspective situated in a specific space. I interchangeably use Indigenous and Anishinabek as, at a global scale, people are familiar with Indigenous, but will understand that Anishinabek is one Nation of the many Indigenous Nations.

Anishinabek ontology-epistemology has protocols based on minobimadziwin (the good life). "Indigenous research is often guided by the knowledge found within. Aboriginal epistemology (the ways of knowing our reality) honors our inner being as the place where Spirit lives, our dreams reside, and our heart beats" [8]. In following Anishianbek protocols guided by the spirit within, I went to St Mary's river making an offering and asking, what is nibi governance, and how can I articulate to others my understanding of nibi governance based on Anishinabek giikendaaswin. I was reminded of the numerous ways of acquiring giikendaaswin: direct observation and experiential learning; storytelling; ceremony; learning from Elders; and vision and dreams. This re-affirmed that giikendaaswin is a "lived knowledge" and cannot be separated from "human experience and action" [19–22]. Peacock (2013) stated that "oral stories are among humankind's oldest way of teaching" [20]; thus, I look at the stories as a source of giikendaaswin for this paper. This provided me with the reminder to utilize several giikendaaswin sources, because learning from ceremony also connects us to our ancestors and spirit helpers providing "opportunities to become seekers of sacred and traditional knowledge" [23], and providing an offering is ceremony. Elders are those that acquired wisdom and are primary teachers of giikendaaswin [21], and I can utilize their giikendaaswin when

speaking for nibi. Geniusz (2009) and Johnston (1982) explained that Anishinabek peoples sought vision for giikendaaswin and guidance throughout their lives [22,24] and, when I made the offering, this was a form of seeking guidance and giikendaaswin. However, Absolon (2011) warned that, often, Anishinabek methodologies are not perceived as valid sources of knowledge within the western science world and, therefore, may not be taken seriously [8]. To counter the possibility of not being taken seriously, books, literature, and journal articles are used throughout as sources of giikendaaswin. This allows me to be in two worlds gathering giikendaaswin from all sources, providing my insight into nibi ontology-epistemology and nibi governance.

## 3. Anishinabe Women's Role in Water Governance

Men and women often engage in different activities utilizing different relatives (known as resources in the colonial context) and, therefore, have specific giikendaaswin. Anishinabek women have a special relationship with the waters since women have life-giving powers [25]. Kermoal and Altamirano-Jimenez (2016) state that women could provide a unique and valuable perspective in the water crisis [1]. Women have a specific relationship with and responsibility to nibi.

I participated in several Mother Earth Water Walks and was the co-lead for the Four Directions Water Walk for the northern direction in 2011. The Water Walks are led by Anishinabe Grandmother Josephine Mandamin with the intent of teaching others the importance of nibi. The Walks began with Lake Superior in 2003 and covered each Great Lake and St Lawrence, routinely covering distances of over 1000 km [7]. The goals of the Walks are to raise awareness of nibi and to change the perception of nibi as resources to spiritual entities. This movement is led by women who are fulfilling their roles to and responsibilities with the waters by engaging people to raise awareness of the spiritual and cultural significance of the waters [7]. These Walks inspired many other Anishinabek women in different communities who coordinated similar events of their own in their communities, re-establishing their relationships with and responsibilities to nibi.

Through the Water Walks and nibi ceremonies I gained a better understanding of my relationship and responsibilities to nibi as an Anishinabe qwe. I also learned of my relationship with nibi in hosting nibi gatherings for the Chiefs of Ontario. It was explained how women have the primary role in responsibilities to nibi due to our relationship of carrying a child in water in our wombs. Cave and McKay (2016) stated that "Indigenous women share a sacred connection to the spirit of the water through their role as child-bearers and have particular responsibilities to protect and nurture water" [26]. Anderson (2010) reiterated women's responsibilities to nibi and that women are known to be "carriers of water" [27]. The Water Declaration of the Anishinabek, Mushegowuk, and Onkwehonwe in Ontario states that "the Anishinabek, Mushegowuk, and Onkwehonwe women are keepers of the waters, as women bring babies into the world carried on the breaking of the water" [10]. The Water Walks raised this awareness of women's special relationship with nibi and responsibilities to nibi. Craft (2014) and Anderson (2010) quoted Elders who reiterate that women have a special connection to the waters as women have the ability to give birth through the waters, and McGregor (2001) also reiterated the responsibility women have to the waters by sharing information about the "Akii Kwe: Anishinabe women who speak for the water" [6,27,28]. The Chiefs of Ontario Report on the First Nations Water Policy Forum (2008) has several quotes from participants stating that women have a special responsibility to nibi and are the "water keepers" [29]. There is no denying women's special relationships with nibi. This special relationship that women have with nibi needs to be transmitted to the world, as more women are picking up their responsibilities to and re-establishing their relationships with nibi, with regards to how nibi governance can be transformed to better protect nibi.

The grandmothers in Anderson's (2010) paper reiterate the relationship and the connections to the sky world, and discuss women's responsibilities for the waters [27]. Anderson et al. (2013) summarized several interviews with 11 grandmothers sharing knowledge on the waters, including the importance of ceremony reiterating women's special relationship with and responsibilities to the waters [15].

Anishinabek women's giikendaaswin can provide a different insight into nibi decision-making related to responsible nibi governance through relationship-based nibi governance.

Regardless of all the regulations, policies, and agreements made under the current colonial water governance systems, nibi is contaminated. With so many agreements in place, one would think that nibi would be protected, and contamination would cease, but unfortunately that is not the case. I would suggest that part of the problem is the mindset of "water-dependent natural resources", because does this mean that humans are also water-dependent natural resources? Western science views nibi as a natural resource versus the Anishinabek view that water is life with a spirit. Von der Porten et al. (2016) explained how the two different knowledge systems have seemingly different human value systems and comparing them is "almost self-defeating" [13]. From the teachings I learned and from understanding that nibi is life, these two different knowledge systems do not mix, as they are like oil and nibi. It is apparent that there needs to be true collaboration on nibi decision-making with the two mindsets coming together to find feasible solutions for managing the inappropriate human behaviors toward nibi, including the need to have Anishinbek women contributing to nibi management policies in nibi governance.

## 4. What is Nibi (Water)?

The Oxford Dictionary (2004) defines water as "a colorless transparent odorless tasteless liquid compound of oxygen and hydrogen" [30]. Settler cultures are less attentive to the relationship between humans and water, and mainstream society typically sees water as a resource or a commodity [31,32], in stark contrast to its treatment in Anishinabek nibi governance. Linton (2010) explained that "$H_2O$ consists of an oxide of hydrogen $H_2O$ or $(H_2O)_x$ in the proportion of two atoms of hydrogen to one atom of oxygen, and is an odorless, tasteless . . . " [33]. Linton (2010) also discussed "modern water" as the dominant way of knowing and relating to water, made known as an abstract measurable quantity by reducing it to a unit—$H_2O$ [33]. These definitions and understandings flow against Anishinabe giikendaaswin, as nibi is alive with responsibilities to life. This is the basic difference between colonial ontology and Anishinabe ontology-epistemology as Blackstock (2001) explained: "water is a meditative medium, a purifier, a source of power, and most importantly has a spirit" [34]. McGregor (2001) reiterated that "water is life" and is considered "a living-entity" [6]. Several articles also stated that water is life, water is sacred, and water is alive with a spirit [19,26–29,34–40]. This difference in understanding what nibi is is the basis of how different peoples manage, understand, and exist with nibi. Yazzie and Baldy (2018) reiterated that water is not a resource to be used by corporations but is a relative [11]. Anishinabek peoples believe that we humans are the baby of Creation with all other beings being alive and our teachers and relatives. In ceremony, I am often reminded that we are all related and everything is alive. There is a need for regenerating the waters' relations, confirming that water is a relative to Indigenous peoples [39] and is alive with a spirit. The colonial understanding of what "water" is is opposite to what Anishinabek understand in allowing nibi to manage itself as opposed to the arrogance of thinking humans can manage other beings such as nibi.

The degradation of water around the world is a pressing issue. Pal Kaur from India in his letter to the Water Voices from around the World stated that water is threatened due to industrialization, urbanization, and widespread deforestation (as cited in Marks, 2007). High Chief Vaasiliifiti Moelagi Jackson of Samoa in his letter to the Water Voices from around the World shared his view on the destruction of the waters with a specific focus on water issues on his island (as cited in Marks, 2007). The TEK Elders along the North Shore of Lake Huron stated numerous times over several years that nibi is contaminated. The United Nations recognized the conditions of the waters by declaring an "international decade of action in 2005" [39]. We know the conditions of nibi, we know the conditions are based on human activities, and we certainly have the ability to change our behaviors toward nibi, but the question remains whether we are willing to change in time to allow nibi to heal, allowing nibi to govern itself by living its responsibility in supporting all life.

Anishinaabek peoples have unique relationships with and understandings of nibi. Anishinaabek peoples are often the first to take notice of the degradation of the waters, and the first to suffer from it due to their close relationships with the waters [6,28,29]. Many Indigenous worldviews treat the waters as both providing the source for all life and having a spirit, not something to be owned or acquired [28]. During my participation in numerous nibi ceremonies, Mother Earth Water Walks, and nibi gatherings, I heard Elders repeatedly teach about the healing powers of nibi, and women's responsibilities in decision-making for nibi. Academic research corroborates the healing powers of the waters [6,28]. Muru-Lanning (2016) referred to water as a gift with curative powers [41]. This unique relationship with and the understandings of nibi are as old as the Anishinbek peoples' existence on Turtle Island (known as Canada, the United States, and Mexico).

Anishinabek peoples believe nibi has healing powers. Nibi is "an important source of healing" and "during times of difficulty is the time to get healing from the water" [28]. I understand that nibi is regarded as "sacred" and is a "powerful medicine" with "life-giving properties" [23,27,41]. Nibi is simply not just a chemical compound from an Anishinabek ontology. Elders say that, when you are "weighed down with a lot of grief, life is becoming unmanageable, or you are going through a lot of pain, our grandmother and auntie and my mother would say go to the water" [34]. I learned through participating in nibi ceremonies, the Water Walks, and discussing nibi issues with Elders, the Anishinabe protocols and teachings about nibi including how the waters have "curative powers" [27,40], acting as medicine. Blackstock (2001) reinforced that water "heals, inspires, and prophesies" [34]. Understanding that nibi has healing powers will promote more respect for nibi in nibi decision-making.

The waters have knowledge that is passed on as "even water can communicate" [6]. Noori (2013) explained how Elder Andrew Medler explained "how water was a source of information for the Anishinaabeg" [42]. If we can understand how nibi is a source of information, the way we interact with nibi will change. McGregor (2011) also referred to water as a "relative" [25]. Nibi can teach us how to improve decision-making if we choose to listen to it communicating to us. Elders also stated "let it teach us" referring to allowing nibi to teach us [27]. Anderson (2010) shared that Maria Campbell explained how water was a teacher in her life and to look to the water for teachings [27]. If we can view nibi as one of our teachers, we should be able to allow nibi to manage itself.

From an Anishinabe ontology-epistemology, each body of nibi has its own personality and is linked to different nibi spirits. "Bodies of water are considered to have their own unique personalities" [26]. Since Anishinabek giikendaaswin is based on place, collaborating with Ansihinabek peoples is key for sound nibi decision-making. "Each of those Great Lakes we walked around has a reputation, a personality" [43]. Drawing on these can be the basis for utilizing Anishinabek peoples in nibi decision-making. Anishinabek peoples who live near nibi, in the watershed, have knowledge to share that will be more inclusive of nibi decision-making.

Nelson (2013) explained that Anishinabe are "water people" and, since "water is considered a sacred element in life", it must, therefore, be "cherished as an essential relative, elder, and teacher" [44]. Wong (2013) referred to stream knowledge not being lost [45], which can still assist the world in understanding our responsibilities to nibi. Christian (2013) explained the "kin-centric perspective" by Ardith Walkem as the Indigenous perspective of all life being related to one another [45]. These relationships with nibi as nibi people, where nibi is considered sacred, and nibi is a relative, need to be front and center in all that we do, especially managing our behaviors toward nibi.

It is very clear that nibi is life, but the power of nibi as a life-giving force and a life-taking force needs to be understood. Anderson (2010) shared that the "powers of water can be more dangerous, for, according to the grandmothers, water can offer all types of messages and energies" [27]. Elder Mary Louie (cited in Blackstock, 2001) explained that if you do not make offerings to the waters, they can take you. It is more than just understanding nibi power, it is also respecting the power through offerings. I canoe and kayak rivers and lakes and was taught by Elders to always make an offering before being on nibi. This display of respect is reciprocal to the power of nibi which then

provides safe passage. One component of the Water Walks was to make an offering at each body of nibi we walked beside or crossed to show our respect for nibi living their responsibilities providing us with life. It is about being reciprocal in acknowledging nibi governance. Blackstock's (2001) paper explained that you have to communicate with the waters from the heart and that the waters have feelings [34]. Acknowledging nibi through reciprocity by making an offering is understanding that nibi governs itself.

Elder Peter Atkinson (cited in Craft, 2014) explained that the waters have a spirit and are looked after by the spirit. Nelson (2013) explained that "Mishipizhu has always been a guardian of the waters and a keeper of balance between the water spirits, land creatures, and sky beings" [44]. Anderson (2010) discussed the significance of the waters as a spirit and the importance of honoring this spirit [27]. It is our responsibility to honor and feast these spirits for living their responsibilities in caring for nibi. It is about a reciprocal relationship. Anderson (2010) further explained how the relationship with the waters has to do with "kinship, reciprocity, and caregiving ... " [27] and the understanding that the waters have a spirit can lead to "how relationships can be established between water and other entities" [27]. These responsibilities to nibi do not rest on Anishinabek peoples alone. Elder Mary Thomas in Blackstock's (2001) work stated "You cannot live without the waters—your body is two-thirds water" [34]. Every human and being on this planet requires nibi to survive, so it is incumbent as humans to acknowledge this relationship and live our responsibility of gratitude to nibi for life. Regional Chief Angus Toulouse in his closing remarks at the First Nations Water Policy Forum documented in the Report reminded everyone that "the future of water is our most precious gift that sustains us all—Indigenous and settlers ... " [29]. The book Water Voices from Around the World has letters from peoples from all around the world about the waters and many declare that the waters are a responsibility of all people [39]. It has always been every human's responsibility to respect nibi by not destroying it. Collaboration with all peoples is needed to produce a reformed understanding of Anishinabek ontology-epistemology.

## 5. What is Anishinabek Ontology-Epistemology?

Searching for giikendaaswin is not a new concept to Anishinaabe peoples. Johnson (2010) illuminated that, traditionally, the older generation passed on its knowledge to the younger generation in many different forms [46] and that stories were "primary vessels of knowledge" [47]. Anishinabek ontology-epistemology is searching for giikendaaswin and can only come through the nuance of relationships and responsibilities. To be Anishinabek is a way of searching, a way of living, and is "deeply embedded in narrative acts of intention, perspective, and community making" [48]. Since epistemology is the study of knowledge, Meyer (2003) explained that Hawaiian epistemology is the cultural or traditional practice of knowledge based on sources found in her territory [18]. Hawaiian people are Indigenous to their territory and, thus, have the knowledge of their territory based on their culture and traditional practices including knowledge passed through generations from living on the lands and waters. This is the same for Anishinabek peoples, which means ontology-epistemology becomes the knowledge of place by a peoples' living on the lands and having relationships with and responsibilities to everything around them, including nibi. Meyer (2003) confirmed this by stating the idea of lifestyles, place, home, people, visions, and lands educates, which opens a doorway to consciousness and awareness of who people are [18].

Anishinabek giikendaaswin is considered a gift based on "reciprocity, respect, relationships, responsibility, and reflection" [5,9,21,48]. Re-searching for gikendaaswin is living ontology-epistemology through various means of Anishinabek acquisition. Anishinabek acquisition comes in many forms and arises from "multiple sources, such as direct observation, experiential learning, learning from Elders, storytelling, ceremonies, contact with non-human entities, and through visions and dreams" [5,22]. I was told that akinomaage means to draw giikendaaswin from all that is around us—the sun, the moon, the animals, the waves, the winds, the plants, and the stars. Waindubence explained at a sunrise ceremony that giikendaaswin comes from the land, stories,

and ceremony, and is based on love and compassion. He explained that your name and clan provide giikendaaswin, but it is about human behaviors, relationships, laws, and actions; it is all encompassing of life and life's experiences. Nbwaakaawin is the word for wisdom, as it takes nbwaakaawin and zaagidwin (love) to apply giikendaaswin. "When you listen, you become aware. That is for your head. When you hear, you awaken. That is for your heart. When you feel, it becomes part of you. That is for your spirit . . . it is so you learn to listen with your whole being. That is how you learn" [49]. Again, the understanding of "doing" is what we learn from; therefore, one must participate [18] and to turn giikendaaswin for nbwaakaawin, the heart is needed. Zaagidwin is love from the heart, meaning giikendaaswin can give true power when one learns to truly listen, participate, and act, and this is nbwaakaawin. This power gained is not based on colonial mindsets of gain and control, it is based on Anishinabek principles of minobimaadziwin (living a good life). Kermoal and Altamirano-Jimenez (2016) stated that Indigenous knowledge "is not fragmented into silos or categories; rather, ontologies, epistemologies, and experiences are interwoven into this system" [1] with minobimaadziwin being the goal. Anishinabek ontology-epistemology cannot be separated.

Searching for giikendaaswin is "full body, full mind"—it is the spiritual, physical, mental, and social [18]. To search is studying and learning, and is commonly coined as research in western academia. Re-search or searching is not new to Anishinabek peoples as Elders speak of the lifelong journey of learning. Absolon (2011) reiterated that "Indigenous peoples have always had means of seeking and accessing knowledge" [8]. Elder Linda Toulouse explained to me that Ndod-ne-aah-non chi-kendaaswin (I am searching for knowledge) is what all Anishinabek did and many continue to do. Dumont (2006) explained in Indigenous Intelligence that re-searching is more than just seeking giikendaaswin, it is "the intelligence of the mind, the intelligence of the heart, the intelligence of the body, the intelligence of the soul, and the intelligence of the spirit" [50]. Colonialism may have changed how we are taught giikendaaswin, how we understand giikendaaswin, how we share giikendaaswin, and how we experience giikendaaswin, but with Anishinabek re-searching their responsibilities, listening to Elders, participating in ceremonies, and the resilience of Anishinabek peoples, people are "returning to ourselves" to find giikendaaswin [22]. Anishinabek ontology-epistemology is all encompassing, reaching far beyond the typical colonial way of obtaining giikendaaswin through academic institutions. Anishinabek research methods are numerous, being as old as the peoples themselves.

## 6. What Is Water Governance?

Nibi governance and nibi-related decision-making is not a new concept to Anishinabek peoples, including Anishinabek women. Anishinabek Elders state that we are the waters, water is life, and the waters have a spirit [7,23,27,28,34]. Anishinabek governance is based on relationships with and responsibilities to all life which includes nibi governance. Elders state that there is "no separation between water and human beings, as we are water, and water is us; if we respect water, we are respecting ourselves" [23]. If we harm the waters, we harm ourselves. This is the basis of nibi governance from Anishinabe ontology-epistemology.

When you ask an Elder what nibi governance is, they will most likely look at you strangely, chuckle, and say I do not know. When you explain that nibi governance is "generally referred to as decision-making processes through which water is managed" [25], they will smile and provide an answer typically beginning with nibi is life. In Blackstock's (2001) article, the Elders reiterate by explaining what water is from an Indigenous perspective including the spiritual perspective [34]. They will talk about the conditions of nibi, how this affects their health and why this happened. In Joe's (2012) model for water governance, the Elders discuss the value of a watershed, cultural water use, principles guiding water use, changes to the waters, and key concerns [37]. Nibi governance to Anishinabek peoples is all encompassing, considering how nibi is treated, including the spiritual relationships and how nibi should be treated, i.e., relationship governance.

Many Elders discuss the other beings that are part of nibi governance as they are responsible for nibi. Blackstock (2001) explained that the waters have a strong spirit, which can be gentle or powerful, forgiving or angry [2]. In ceremony, we are told that everything, all life has a responsibility to each other. McGregor (2012) explained how the water purifies the earth in the re-Creation story as people are not living their responsibilities, and harmony and balance are disrupting how the water-beings play an integral role in restoring life [3]. Maintaining balance and harmony through good relationships is nibi governance.

In all the conversations, articles, and books I read with a specific focus on nibi, the theme of how humans behave is dominant in how humans make decisions to govern nibi. Nibi already knows how to govern itself, and it lives its responsibilities to life; it is about how humans govern their relationships and selves to nibi that should the primary focus. McGregor described the Indigenous realities in relation to the waters and concluded that "water is a living spiritual being with its own responsibilities to fulfil" [25]. Often in ceremony, the Elders talk about how humans are the baby of Creation with responsibilities to all those who give their lives so we can survive, which means we govern our own behaviors allowing all other beings to govern themselves. Learning to understand and live this is wisdom and is being humble in knowing you can and should only govern yourself.

## 7. Implications of Nibi Governance

Control, pollution, and dispossession of lands and nibi were common tactics used by colonization to conquer Anishinabe peoples. Bakker et al. (2018) explained that creating reserves was a strategy to remove Indigenous peoples from the lands [12]. Absolon (2011) stated that "colonialism attempts to eradicate every aspect of who we are" [8], which also means an attack on our giikendaaswin systems by aiming to integrate Anishinabek into mainstream society. These attempts have dire consequences for Anishinabek peoples, as many have forgotten who they are; however, many Anishinabek peoples are returning to the lodge, learning from our Ancestors, the lands, the ceremonies, and the Elders, providing resurgence in understanding who we are, what our responsibilities are, what are relationships mean. This resurgence is providing opportunities for collaboration on nibi governance by utilizing knowledge systems from two worldviews.

The Chiefs of Ontario coordinated several meetings in relation to nibi with First Nation Elders, technical staff, and Chiefs, and the outcome of these meetings was the Water Declaration of the Anishinabe, Muskegowuk, and Onkwehonwe in Ontario. There is a section in the Declaration dedicated to the conditions of nibi with 11 points reiterating that nibi is contaminated and all life is affected by this contamination [10]. The Declaration also states that the contamination of nibi is due to the "intervention of non-Indigenous people" [10]. This intervention on nibi is not allowing nibi to live its responsibilities. It is obvious there are two different mindsets about nibi. Indigenous peoples all over the world believe water is life. The colonial mindset believes nibi is a resource to be used and wasted based on human needs alone. In order to stop the destruction of nibi, it is necessary for current nibi decision-makers to stop making decisions based on the colonial mindset and to ask Anishinabek women what needs to be done.

Christian (2013) discussed how it is imperative for all people to literally track the waters, as this is what will bring us to an understanding of how each of us is connected to one another and to the waters to which we belong [45]. Anderson (2010) shared the grandmothers' statements that "we are water, and we need water to stay alive" [27]. It is apparent that all life depends on nibi, but nibi does not depend on human existence. If humans were eradicated from this Earth, all life would continue to function and live their responsibilities; if nibi disappeared or simply stopped living its responsibilities, human life would perish. Reconnecting to nibi provides a better understanding that we are all connected to nibi, allowing for a better understanding of how to share nibi with all life.

The current government regulations and laws allow nibi to be poisoned, which is in direct conflict with Anishinabek Nibi giikendaaswin. Bakker et al. (2018) explained that regulatory injustices affecting water security for Indigenous peoples is due to competing jurisdictional priorities between different

levels of governments [12]. Elders shared stories of water contamination and how this affected the health of their peoples [27]. Anishinabek women are willing to share their giikendaaswin based on their relationship with and responsibilities to nibi to find solutions to the current nibi crisis. Anderson et al. (2013) provided a solution to the current water management strategies by stating "understanding Aboriginal women's perspectives is critical in the formulation of water management strategies . . . because women are considered the holders of 'water knowledge' and assume a primary role in the protection of water resources" [15]. McGregor's section of the Chiefs of Ontario Submission to the Walkerton Inquiry discussed the spirituality of water, the importance of Aboriginal water knowledge, and how the colonial system oppressed Aboriginal knowledge through broken treaties, environmental injustices, residential schools, and dominant policies; she provided examples of Anishinabe women who speak for the waters and how Aboriginal peoples are willing to share their knowledge as solutions to reform water management policies and strategies [6]. Muru-Lanning (2016) outlined various ways for a new system of water management which respects the Indigenous peoples of the area and provides an understanding of how the waters are ancestral [40]. Barlow, in her letter to Water Voices from Around the World, discussed that "water will be nature's gift to humanity to show us the ways of peace" (as cited in Marks, 2007, p. 29). Scholars more recently discussed what meaningful collaboration for water governance looks like by identifying opportunities for reform and water justice, and offering practical solutions to inform collaborative water governance [12–14,36]. Telling governments and nibi decision-makers to utilize Anishinbek women's giikendaaswin is not new, the question becomes when will the space be created to collaborate on nibi governance?

Nibi can govern itself; it is the human behavior toward nibi that needs to be governed. I have come to understand from teachings and ceremony that nibi will continue to flow and live its responsibilities in governing itself. Anderson (2010) shared the grandmothers' statements that "we are water, and we need water to stay alive" [27]. With humans relying on nibi, there is a demand to refocus our understanding of human's responsibilities to and relationships with nibi. In ceremony, we are told that everything out there knows its responsibilities; it is us humans that have forgotten our responsibilities; however, as we return to the lodge, our responsibilities are being picked up and acted upon. Our relationships with nibi are being renewed, which provides opportunities for collaboration and better water decision-making.

## 8. Conclusions

It is obvious there are two different ontologies and epistemologies, i.e., worldviews, toward nibi and nibi governance. Indigenous peoples all over the world believe water is life. The colonial worldview is based on water being a resource to be used and managed based on their human needs alone. McGregor (2001) stated that the waters are "viewed as a natural resource rather than a gift from the Creator" [6]. The colonial mindset allows nibi to be poisoned with the arrogance that it is infinite and can be treated with another poison so that it is fit for human consumption. The Water Declaration of the Anishinaabek, Mushkegowuk, and Onkwehonwe in Ontario has a section on the conditions of the waters, stating how the waters are poisoned by "foreign economic values" violating the "sacred laws given by the Creator" [10]. In fact, there are several Declarations from around the world that asserted Indigenous views and giikdendaaswin about the waters, all referring to the immediate need for Indigenous peoples and their giikendaaswin systems to be included in water governance. McGregor (2011) discussed how Anishinabek peoples regard western water management approaches to be lacking by themselves to address the challenges the world, countries, cities, and communities are facing [25]. Anishinabek peoples provided Nibi Declarations, scholarly articles, and actions to express their concerns on the lack of nibi decision-making and the state of nibi; however, is the non-Indigenous world ready to read, to understand, listen, and participate to truly stop the destructive behaviors toward nibi? Will a space be created to collaborate?

Anishinabek peoples sustained themselves and nibi for thousands of years. Travers (2016) reiterated that Anishinabek peoples had sound stewardship over the Great Lakes [51]. There is a need for respect for different cultures, to be mindful of our actions toward the waters, and to

rebuild and acknowledge the Indigenous perspectives on the waters [44]. Christian explained what cultural interface is and the conscious efforts that will be needed to welcome Indigenous people's knowledge providing reform for water decision-making [44]. In too many instances, water governance is primarily based on one dominant culture, not allowing for collaboration from multiple perspectives and knowledge systems. If intellectual space for Anishinabek knowledge is made, opportunities to collaborate on water issues can be solved [1]. Von der Porten (2018) explained that, despite "30 years of grappling with the question of integration", little changed; thus, it might be wise for non-Indigenous environmental practitioners to find ways to support Indigenous peoples in making decisions about their territories [13], which includes nibi decision-making. True, meaningful collaboration with Anishinabek women's knowledge systems will be a new and different way for nibi-related decision-making.

Our relationships need to be reciprocal in all that we do; we need to be grateful and "check our own behavior (not manage the behavior of our relatives through such paradigms as natural resource management)" [51]. Doerfler (2013) explained that we have to "remember our responsibilities to both our ancestors and to future generations; learning about our past and acting accordingly is an act of survivance" [52]. When a person can listen to the lands, this provides "visceral, hands-on, embodied experiences of a reality not made by human thought"; some people refer to these experiences as "ecoliteracy" [52]. Nibi governance for Anishinabek peoples is about listening to nibi, and living our responsibilities to and maintaining our relationships with nibi, which provides another way for nibi governance.

Anishinabek women have specific giikendaaswin stemming from their relationships with and responsibilities to nibi. Anishinabek women have special connections to nibi through their roles as child-bearers, and the giikendaaswin that Anishinabek women have can provide a different insight into nibi decision-making relating to responsible nibi governance through a governance of our relationship with nibi. Regardless of the attempts to erase this giikendaaswin, Anishinabek women through zaagidwin maintained these relationships and responsibilities and are willing to share to transform nibi governance.

The waters are a common ancestor to all humanity and all life. Maru-Lanning (2016) stated that Tupuna Awa literally means "river with ancestral power" [40]. There is a movement to educate all peoples of nibi as a commons. Barlow (2001) promoted the waters as a commons; "it belongs to everyone and to no one exclusively and must be passed on to future generations in sufficient volume and quality" [31]. Nibi as our common ancestor requires all humanity to re-learn their responsibilities to and re-establish their relationships with nibi by understanding that "water means life" to everything and every being. Rakhmonov in his letter to Water Voices from around the World stated that dealing with the water issue will "benefit people the world over" reiterating the fact that water is needed by all humanity and all humanity must come together to preserve the waters for future generations (as cited in Marks, 2007, p. 9).

In order to re-center the political view in new and different ways in relation to nibi decision-making, uses, and management, Anishinabek women need to be involved in collaboration with other cultures. An educational reform is needed to inform people of the Anishinabek ways of living with nibi, which will contribute to the future of how nibi is regarded and governed. The understanding of Anishinabek ways of living with nibi is old, but will be "new" to many current water decision-making regimes. It is urgent that the "new" understandings of politics and governance in relation to nibi be accepted, as nibi is not infinite; rather, nibi is needed for all life to sustain itself, and we are nibi.

**Funding:** This research received no external funding.

**Acknowledgments:** I want to acknowledge the Anishinabek Elders who have shared their giikendaaswin with me always reminding me that I am on a journey of life-long learning. I also want to thank my son-in-law Paul Mieghan Chiblow for his editing.

**Conflicts of Interest:** The author declares no conflict of interest.

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
