# Peer review of "Anishinabek Women’s Nibi Giikendaaswin (Water Knowledge)"

_water, doi:10.3390/w11020209_

Round 1
Reviewer 1 Report
This is a well-written, reflexive paper on Anishinabe women’s water knowledge. The experiential, reflexive content and style of the paper makes it a contribution in its own right. The author contributes empirical depth, before linking this to broader questions of the role of Anishinabe knowledge in water governance. The arguments have implications for understanding the role of Indigenous knowledges and experiences in water governance more generally, while being attentive to diversity. The argument could nonetheless be clearer and the narrative stronger/more fluid (excuse the pun). It is really only in the conclusion when a lot of the argument becomes clear, but by then some of the arguments seem out of place. So, since I am not in a position to “assess” the empirical content – or indeed much of the conceptual-epistemological content – related to Anishinabe nibi, I will focus on how the discussion is linked to broader issues in water governance. I also have a few points on style. On the use of nibi beyond the Anishinabe context and linking it to broader arguments: E.g. P7, lines 310-313: “In Joe’s (2012) model for nibi governance”. Should the word nibi be used here? Nadia Joe’s report focuses on Aishihik water in the Yukon, yet the use of nibi here implies that she is discussing Anishinabe water. Could you use “…mode for Aishihik water governance…” or something similar instead? There are many other places where nibi is used as water in a general sense (but sometimes you also say ‘water’), rather than Anishinabe water. So there is a degree of slippage here, making it a little bit confusing what is meant by nibi: i.e. is it water in general and you prefer Anishinabe terminology (which is certainly fine, but could be clarified); or is it Anishinabe water specifically (whether conceptual, material, relational, etc.)? Another key example of this, p8 line 394: “Indigenous peoples all over the world believe nibi is life”. Here you are using nibi in an abstract, universalized sense (ironically, a bit like Linton’s “modern water”). I think this is risky, as you risk readers interpreting that you are suggesting that all of the relational components related to Anishinabe nibi that you highlight can be applied to water across Indigenous populations. Elsewhere you state that the perspective you present is contextual, according to Anishinabe experience. So I think the paper needs an explanation of the links you are making between nibi (governance) and water (governance) more generally – if indeed you are making such a linked argument. Another example is stating that to stop the destruction of nibi, Anishinabe women need be decision makers: the is true among Anishinabe but the argument is a little confusing because you have also been using nibi to refer to water more broadly (and surely Anishinabe women shouldn’t be making decisions about tsilhqot’in waters, for example). Relatedly, there is a risk in the paper of creating a false dichotomy between Indigenous water and Colonial water (as in, suggesting that these are the only two waters, even as they may contain diversity and even as they are certainly opposed). First, as per the comments above, not all Indigenous water is nibi. Second, not all indigenous water can be reduced to the statement that water is alive. There are many diverse relational engagements that provide such life, and I’m sure the author doesn’t mean to gloss over them. Third, not all non-indigenous water should be reduced to colonial water. Fourth, not all colonial waters entail conceiving water only as a resource. Fifth, not all colonial waters as a resource conceive water as not alive (i.e. there might be a false opposition being drawn between water as alive and colonial water as dead). Sixth, there may be other waters that are not entirely indigenous, nor entirely colonial (without suggesting some kind of hybrid, it seems somewhat reductionist to imply that these are the only two worldviews). I’m not suggesting that you shouldn’t draw attention to the destructive aspects of colonial waters particularly on Indigenous water; but the way in which this is done could use more explanation and nuance. On strengthening the argument: Overall, the paper reads as somewhat descriptive of Anishinabe nibi and eventually Anishinabe nibi giikendaaswin and the role of women in nibi governance. I wonder if there is a more analytical way of presenting the material. For example, towards the end, the argument is made that different concepts, epistemologies, or perhaps ontologies of water “do not mesh” (p8). Is there a way of getting to this argument thematically? While there are times when Anishinabe (and indeed sometimes Indigenous) knowledges are contrasted with Colonial knowledges. But perhaps a focus on the impact that colonial knowledge systems have had on Anishinabe knowledge generation and sharing would help to tease out an answer to why they do not mesh. Relatedly, this might help to unpack what “true collaboration” means in “environmental decision-making”, which is not really unpacked in the paper, despite pointing to the potential of reciprocity. In fact, I would suggest adding a section before the conclusion that discusses the implications of the paper up to that point (some content from the conclusion could go here, since it is quite long). This section might address specifically the political question of water governance, given what has been presented about Anishinabe nibi and Anishinabe nibi giikendaaswin. This might be the place to address the above issues about making links between nibi, diversity among indigenous waters, and water more generally (colonial or otherwise). At the end of the conclusion you make some fairly large statements about humanity’s relationship with water. But the link from Anishinabe nibi to this global water seems to be missing, and might not be sustainable. In sum, I think there are good arguments that can be made in this paper that do go beyond Anishinabe nibi. Currently, however, the links are missing and a little more care is required to clarify concepts, their application, and their link to any broader argument. Style and other minor points: - At times the paper begins to read a little like an interconnected set of bullet points; such as “author X states this; author Y states that”. There is nothing particularly wrong with this, but I think there is room for a more engaging style if possible, especially given the evocative content at times. - There are some errors in syntax or incompletely conjugated sentences, so a thorough proof-read would be useful. Here is just once example from lines 208-2010, p 5: “It has always been every humans’ responsibility to respect nibi by not destroying it and therefore the need to educate everyone of this responsibility”. Similarly the sentence that follows it is a little confusing: “Collaboration for nibi decision-making is understanding Anishinabek epistemology and ontology”. - In the conclusion (and perhaps earlier) when discussing contamination, you might refer to Leanne Simpson’s work and use notion that water can be contaminated through occupation. In the conclusion, you make a similar argument about foreign economic values and intervention of non-indigenous peoples, for example.Author Response
see attached

Reviewer 2 Report
I think this is an important article making a strong contribution to the growing literature about indigenous experience, knowledge, research and ‘science’ and the challenges of water management. Written from a specific indigenous nation, and indigenous female perspective, the author draws out the important elements of ‘response-ability’ (see Haraway) and deep knowledge that indigenous nations bring to what has become environmental management. This paper provides a useful account of the political actions taken by Anishinabe to exercise responsibilities to support the ongoing wellbeing of the lands, the waters and all living things.
I think the article could be re-structured to strengthen the argument – re-ordering the sections could be useful. I would bring the section ‘4. Indigenous Research Methods and Positionality’ to the beginning of the article and follow this with ‘6. Anishinabe women’s role in water governance’. This would bring the methodology, situation/location and political acts of responsibility to ‘nibi’ to the foreground. The specific relationship, knowledge and responsibility of the author would better situate the detailed discussion of Anishinabe and water. At the end of each section the author makes a strong, clear summary point. I think bringing these together and using them as a guide for tightening the abstract and introduction could be a useful exercise.
I also think the author’s handling of the relationship of (some) western ideas of epistemology and ontology to Anishinabe ways of being would benefit from some engagement with more recent ‘posthuman’ theory. It appears from the article that Anishinabe ‘philosophy’ does not create a ‘division’ between ontology, epistemology and axiology – indigenous scholars have made the point that this is a problematic divide (eg Kyle Powys White). Non-indigenous scholars such as Donna Haraway (Staying with the Trouble) and Karen Barad (philosopher/physicist) may be useful here – in particular Barad’s concept of Ethico-onto-epistem-ology and Haraway’s concept of ‘response-ability’. Referring to the work of these scholars, alongside indigenous scholars would support the author’s discussion of Anishinabe ways of being, philosophy etc.
The paper does have a few grammatical errors and typos – eg line 201 life should be live; Lines 208-2010 ‘It has always been every humans’ responsibility to respect nibi by not destroying it and therefore the need to educate everyone of this responsibility’.
